# Optimization of a natural fermentative medium for submerged mycelial culture of *Ganoderma lucidum* and the nutritional and bioactive composition of the cultured food-grade mycelia

Shizhong Zheng[1,2], Lu Zhang[1,3], Yihui Yang[1,3], Shengrong Liu🔴[1]*, Qi Wei[4], Qianhui Huang[1,2], Weirui Zhang[1,2]

**1** College of Biological Science and Engineering, Ningde Normal University, Ningde, China, **2** Fujian Higher Education Research Center for Local Biological Resources, Ningde, China, **3** College of Life Science, Fujian Agriculture and Forestry University, Fuzhou, China, **4** College of Ocean, Ningde Normal University, Ningde, China

* fjhost@163.com

## Abstract

*Ganoderma lucidum* is a famous traditional Chinese medicinal fungus with health-promoting and pharmacological properties. The present study aimed to provide a liquid medium for *G. lucidum* submerged fermentation suitable for safely utilizing the mycelia and broth and as a solution to potentially improve the process' economy. For this, a natural medium was formulated and optimized. The nutritional and bioactive composition of the mycelia cultivated in the optimized medium and a nutrient-rich (NR) medium were also determined. The one-variable-at-a-time experiments indicated that among various tested carbon sources, sucrose was favorable for mycelial growth. The supplementation of corn flour in the medium promoted mycelial growth and its optimal level was 7.0 g/L. A further study by response surface methodology obtained an optimal medium comprising of 300 g/L potato (use its extract), 25 g/L sucrose, and 7.5 g/L corn flour. Using the optimized medium, the mycelial yield reached 7.51 g/L. The crude ash (10.43%) and fiber (10.51%) content of the mycelia cultivated in the optimal medium was higher than those (ash 5.75% and fiber 5.93%) of the mycelia grown in the NR medium, while the proteins content and all examined amino acids were lower than those of the mycelia cultivated in the NR medium. The mycelia contained 32.24 mg/g triterpenoids and 3.42% intracellular polysaccharides, which is higher than those (triterpenoids 20.35 mg/g and intracellular polysaccharides 2.03%) of the mycelia cultivated with the NR medium. These findings indicated that the cultured food-grade mycelia had the potential for use in the pharmacological industry. Meanwhile, the broth had the potential for safe use in foods due to free of chemicals. The present study provides a reference for improving the economy of *G. lucidum* submerged culture by using a natural medium, and its industrial fermentation may be economically viable.

**Data availability statement:** All relevant data are within the manuscript and its Supporting Information files.

**Funding:** This work was supported by the financial support of Fujian Provincial Department of Science and Technology (2021I0046). The funders had no role in study design, data collection and analysis, decision to publish, or preparation of the manuscript.

**Competing interests:** The authors have declared that no competing interests exist.

## Introduction

*Ganoderma lucidum* (Fr.) Karst belonging to the family Polyporaceae, commonly known as Lingzhi in Chinese, is a famous medicinal fungus that is widely used to prevent and treat various diseases such as asthma, bronchitis, hepatitis, hypertension, hepatopathy, arthritis, nephritis, gastric ulcer and cancers in oriental countries for centuries [1]. *G. lucidum* possesses diverse biological and pharmacological activities, including antitumor, hypotensive, hypoglycemic, immune modulatory, anti-inflammatory, skin lightening, neuroprotective, and anti-oxidative effects [1,2]. In the fruit bodies, spores, and mycelia of *G. lucidum,* a wide range of physiologically active substances have been isolated. These include polysaccharides, triterpenoids, steroids, alkaloids, and nucleotides [3,4]. Among these, the polysaccharides and triterpenoids are the two most important bioactive constituents in this fungus.

To meet the great demand for high-quality *G. lucidum* products in market, its artificial solid cultivation on short trunks or synthetic substrates is widely adopted in the production of fruiting bodies. With the fruiting bodies and spores as raw materials, a wide variety of *G. lucidum* products such as slices, powder, tablets, capsule, granule, and injections have been developed and manufactured [5]. Presently, the artificial solid cultivation of *G. lucidum* is prosperous in China, Korea, India, and several Southeast Asian countries. Herein, it should be pointed out that the solid cultivation of *G. lucidum* has several disadvantages, including long cropping cycle, unstable product quality, requirement of a large space, and a high influence of environmental changes [6,7].

Submerged culture of mushrooms has the potential of high biomass yield in a compact space over a short time with less risk of contamination [8]. Hence, it has been viewed as a promising alternate to solid cultures for the production of mushrooms in the form of mycelia and bioactive metabolites. As *G. lucidum* has perceived health-promoting effects and high medicinal value, its submerged culture has received considerable attention, mainly for the production of mycelia, polysaccharides and triterpenoids. To improve the productivity, several strategies such as optimization of medium and culture conditions [9,10], development of novel processes [11,12], elicitation by addition of inducers [13], genetic manipulation of strains [14,15], and utilization of solid seed [16] have been explored, and significant improvements have been achieved. From a viewpoint of industrial application, the productivity of both mycelia and bioactive metabolites are still low, making the submerged fermentation process of *G. lucidum* industrially unattractive. Another obstacle is that there are many technical and engineering problems associated with *G. lucidum* submerged fermentation in bioreactors [17]. Under these circumstances, the effective utilization of all the fermentation products including mycelia and culture broth may be a reasonable solution to improve the bioprocess' economy.

The objective of the present study was to provide a desirable medium for submerged mycelial culture of *G. lucidum* suitable for the safe utilization of its fermentation products (mycelia and culture broth). For this purpose, a food-grade medium was typically formulated using agricultural products and food ingredients, and optimized

using the empirical one-variable-at-a-time approach and response surface methodology. Furthermore, in order to provide a scientific basis for better utilizing the cultured food-grade mycelia, the content of nutritional and bioactive composition was investigated. The present study provides a new perspective to perform *G. lucidum* submerged fermentation by using a natural medium for the safe utilization of all the fermentation products, thus could potentially improve the economy of process, making its large-scale industrial fermentation economically feasible.

## Materials and methods

### Mushroom strain

*Ganoderma lucidum* G10016, obtained from the Mycological Research Centre of Fujian Agriculture and Forestry University (Fuzhou, China), was used throughout this study. It was incubated at 25°C for 7 days on potato dextrose agar (PDA) slants, stored at 4°C, and sub-cultured every four weeks.

### Media

PDA medium comprised of 200 g potato, 20 g glucose, and 18 g agar in 1 L distilled water. Potato dextrose broth (PDB) was PDA medium by omission of the agar. For comparison, a nutrient-rich (NR) medium was used, which was comprised of 30 g glucose, 5 g peptone, 4 g yeast extract, 1 g $KH_2PO_4$, and 0.5 g $MgSO_4$ in 1 L distilled water.

### Inoculum preparation

Ten mycelial agar discs (7 mm in diameter), cut from the colony edge of the actively growing cultures in PDA plates, were inoculated into a 250 mL Erlenmeyer flask with 50 mL of PDB medium, and shaken in a rotary incubator at 150 rpm and 25°C for 5 days. Then, the culture was homogenized with an ultrasonic cell disrupting apparatus (SCIENTZ-IID, Ningbo Scientz Biotechnology, Co., Ltd., Ningbo, China) with power input of 550 W and frequency of 20 kHz using a 6 mm diameter horn-type probe. The homogenate (5 mL) was transferred to 50 mL PDB in a 250 mL flask, and cultivated further for 2 days. The finally obtained mycelial culture was used as liquid seed.

### Submerged culture

All submerged cultures in this study were performed in 250 mL Erlenmeyer flasks with 80 mL medium. The inoculation level of liquid seed was all set at 10% (v/v). The flasks were cultivated at 25°C and 150 rpm for 4 days. The samples were taken for estimating mycelial biomass.

### One-variable-at-one-time experimental design

PDB medium was used as basic medium in *G. lucidum* submerged cultures. To select a suitable carbon source for mycelial growth, the glucose in the basic medium was replaced individually with sucrose, fructose, starch, and maltose, each at 20 g/L. At the end of submerged cultivation, the biomass production was measured, and then a desirable carbon source was selected. After the screening, the effect of a low-cost, easily available ingredient corn flour on the mycelial growth was tested in the range of 1.0 to 11.0 g/L, with the selected carbon source in the medium, and its optimal level was determined based on biomass yield.

### Response surface experimental design

According to the results of one-variable-at-a-time experiment, three medium compositions, i.e., potato, sucrose, and corn flour were considered as independent variables, and their levels in the media were optimized using response surface methodology (RSM) with Box-Behnken design (BBD) for maximizing mycelial production. Design-Expert version 8.0 was used for experiment design, data analysis, mapping, and optimization.

The symbols, the actual and coded values of the independent variables are shown in Table 1. Each variable was tested at three levels, coded as low (−1), medium (0), and high (+1) levels, respectively.

The matrix of BBD is shown in Table 2. The design consisted of a total of 17 runs, with three replicates at the center point to estimate the pure error of the experiments. The submerged culture experiments were performed as previously described. The biomass yield was taken as the response variable. To correlate the response variable to the independent variables, the results were fitted into the following second-order polynomial equation using a multiple regression technique.

$$Y = \beta_0 + \sum_{i=1}^{n} \beta_i X_i + \sum_{i=1}^{n} \beta_{ii} X_i + \sum_{\substack{i=1 \\ j>i}}^{n-1} \sum_{j=2}^{n} \beta_{ij} X_i X_j + \varepsilon$$

**Table 1. Range and levels of the independent variables investigated in the Box-Behnken design (BBD) for biomass production of *G. lucidum* in submerged culture conditions.**

| Independent variables | Symbol | | Range and levels | | |
|---|---|---|---|---|---|
| | Uncoded | Coded | −1 | 0 | 1 |
| Potato (g/L) | $x_1$ | $X_1$ | 100 | 200 | 300 |
| Sucrose (g/L) | $x_2$ | $X_2$ | 15 | 25 | 35 |
| Corn flour (g/L) | $x_3$ | $X_3$ | 2.5 | 5 | 7.5 |

$X_1 = (x_1-100)/100$, $X_2 = (x_2-25)/10$, and $X_3 = (x_3-5)/2.5$.

**Table 2. The matrix of BBD with coded and actual (in parentheses) values of the three independent variables along with results of biomass production.**

| Run | Independent variables | | | Biomass (g/L) |
|---|---|---|---|---|
| | $X_1$ (potato) | $X_2$ (sucrose) | $X_3$ (corn flour) | |
| 1 | −1 (100) | 0 (25) | −1 (2.5) | 3.76±0.13 |
| 2 | 0 (200) | 0 (25) | 0 (5.0) | 5.86±0.15 |
| 3 | 0 (200) | 1 (35) | −1 (2.5) | 4.06±0.21 |
| 4 | 0 (300) | −1 (15) | 0 (5.0) | 5.04±0.09 |
| 5 | −1 (100) | 1 (35) | 0 (5.0) | 4.50±0.17 |
| 6 | 1 (300) | 0 (25) | 1 (7.5) | 7.26±0.24 |
| 7 | 0 (200) | −1 (15) | 1 (7.5) | 6.44±0.14 |
| 8 | 0 (200) | 0 (25) | 0 (5.0) | 6.10±0.22 |
| 9 | 0 (200) | 0 (25) | 0 (5.0) | 6.22±0.18 |
| 10 | 0 (200) | 1 (35) | 1 (7.5) | 6.50±0.14 |
| 11 | 1 (300) | 1 (35) | 0 (5.0) | 7.04±0.18 |
| 12 | −1 (100) | 0 (25) | 1 (7.5) | 5.48±0.15 |
| 13 | 0 (200) | 0 (25) | 0 (5.0) | 6.24±0.11 |
| 14 | −1 (100) | −1 (15) | 0 (5.0) | 4.50±0.21 |
| 15 | 0 (200) | 0 (25) | 0 (5.0) | 5.76±0.26 |
| 16 | 1 (300) | 0 (25) | −1 (2.5) | 4.74±0.17 |
| 17 | 0 (200) | −1 (15) | −1 (2.5) | 4.08±0.20 |

where $Y$ is the predicted biomass yield, $\beta_0$ is the model intercept term, and $\beta_j$, $\beta_{jj}$ and $\beta_{ij}$ are the regression coefficients for linear, quadratic, and interaction terms, respectively. $X_i$ and $X_j$ are the coded variables. ε represents the error for the system.

### RSM modeling and statistical checking

The significance of the obtained regression model was determined by an *F*-test, and the analysis of variance (ANOVA) for response surface quadratic model was performed to test the significance of each coefficient in the equation. The fitness of the model was evaluated by the determination coefficient $R^2$, adjusted determination coefficient (Adj $R^2$), and the Lack of Fit *P*-value. The *P* value was used as a tool to check the significance of each of the coefficients and indicate the interaction strength between the independent variables. Values of *P* less than 0.05 were considered statistically significant. Three-dimensional response surfaces and two-dimensional contour plots of the regression equation were drawn to illustrate the relationship between the response and experimental levels of each independent variable and the type of interaction between the two variables.

### Validation of mathematical model

To evaluate the fitted model, validation experiments were carried out with the optimized medium. The difference in biomass yield between the experimentally obtained and predicted values was used to evaluate the accuracy and general availability of the model.

### Fermentative production of mycelia

For the production of *G. lucidum* mycelia, submerged fermentation was carried out in 250 mL flasks with the optimized medium. For comparison, the NR medium was also tested in the fermentation. The mycelial cultures were filtered with gauze and completely washed with distilled water. The clean mycelia were lyophilized, and ground into a fine powder. The mycelial powder was used for the chemical analysis of the nutritional and bioactive compositions.

### Measurement of biomass

To ensure the accurate measurement of mycelial biomass, three independent flask cultures were taken as replicates. All the mycelia in each flask were collected by filtration on gauze, followed by repeated washing with distilled water, and drying at 60°C to a constant weight. The weight of dry mycelia was measured on an electronic balance.

### Proximate composition analysis

The proximate composition of the cultured mycelia of *G. lucidum*, including moisture, crude ash, crude fiber, crude fat, and crude protein, were determined according to methods described by Ouzouni et al. [18]. Ash was determined by combustion in a muffle furnace at 600°C. The protein content was determined with the micro Kjeldahl method, and the nitrogen factor of 4.38 was used for crude protein calculation. The crude fat was determined by extraction with hexane using a Soxhlet system. The total carbohydrate content is calculated by subtracting the percentage contents of ash, fat, and protein from 100%. Amino acid analysis was carried out using an amino acid analyzer after hydrolysis of the mycelia with 6 mol/L HCl for 24 h at 110°C.

### Determination of polysaccharides

To estimate the intracellular polysaccharides, the phenol-sulfuric acid method was used [19]. The powder of mycelia (100 mg) was extracted in 5 mL 1 mol/L NaOH at 60°C for 1 h, and then centrifuged at 8000 × g for 10 min. The resulting supernatants were used in the reaction. A standard curve was also built using D-glucose as the standard, and the content of intracellular polysaccharides in the mycelia was calculated.

## Determination of triterpenoids

For the quantification of total triterpenoids, the vanillin-glacial acetic acid method as described by Cai et al. [20] was used. The dried mycelia were ground in a mortar with pestle and sieved with a 60-mesh stainless screen to obtain a fine powder. The powder (100 mg) was extracted twice in one week, each in 5 mL 80% ethanol. The extracts were combined and centrifuged, and the resulting supernatants were used in the reaction. To calculate the triterpenoids, a standard curve was also built with Ursolic acid as a standard.

## Statistical analysis

All the experiments were carried out in triplicates, and the results were expressed as the mean±standard deviation. Significant differences were determined by Duncan's multiple range tests and Student's *t*-test at *P*< 0.05.

## Results

### Preliminary optimization of medium composition by one-variable-at-a-time approach

**Screening of a suitable carbon source for biomass production.** Fig 1 shows the biomass production of *G. lucidum* in response to different carbon sources in submerged cultures. Among various tested carbon sources, sucrose gave the highest biomass yield of 3.44 g/L, followed by maltose, glucose and fructose with moderate growth, and the least growth of only 1.68 g/L was shown when starch was used. Considering the cost and availability of the carbon sources as well as the biomass yield, sucrose was therefore selected as a favorable carbon source. The results are in agreement with the findings of previous studies in which sucrose was shown to be an excellent carbon source in submerged fermentation of *G. lucidum* for production of biomass and bioactive substances [9,21].

**Effects of corn flour supplementation on biomass production and its optimal level.** With sucrose instead of glucose in PDB medium, the effect of supplementation of inexpensive and easily available corn flour on the mycelial growth of *G. lucidum* was evaluated. As shown in Fig 2, the supplementation of corn flour in the medium greatly enhanced biomass production, and the enhancement effect was closely related to the supplemented level. The maximal biomass

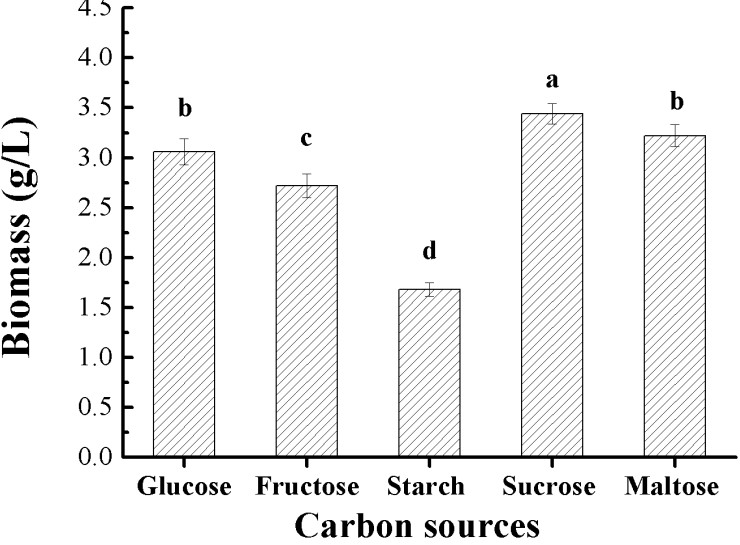

**Fig 1. Effects of different carbon sources on biomass production in *G. lucidum* submerged cultures.** Different letters on the error bars represent significant differences (*P*< 0.05).

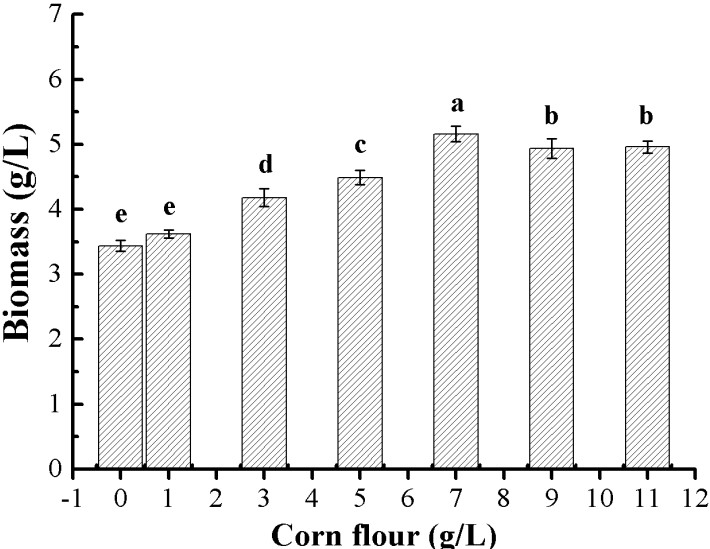

**Fig 2. Effects of corn flour supplementation on biomass production in *G. lucidum* submerged cultures.** Different letters on the error bars represent significant differences ($P < 0.05$).

production attained 5.21 g/L when corn flour was added at 7.0 g/L, 52.61% higher than that (3.44 g/L) obtained in the control medium (without addition). Hence, corn flour was a suitable nutrient composition for promoting biomass production in submerged fermentation of *G. lucidum*. In the literature, corn flour as a carbon source is frequently used in mushroom submerged fermentation, and shown to be effective in promoting mycelial growth and triterpenoids biosynthesis [22,23]. Their findings are generally consistent with our results.

### RSM model fitting and statistical significance

Potato, sucrose, and corn flour were selected as three independent variables in BBD experiments for further optimization by RSM. The matrix of BBD along with the corresponding response values (biomass yield) are shown in Table 2. The biomass production varied significantly among 17 runs, with the minimum biomass yield of 3.76 g/L on Run 1, and a maximum value of 7.26 g/L shown by Run 5. The great difference in biomass yield exhibited that the three studied variables and their interactions had a significant impact on biomass production of *G. lucidum* in submerged cultures.

Thought multiple regression analysis on the experimental results, a fitted second-order polynomial equation with three independent variables in coded values was established as follows:

$$Y = 6.04 + 0.73X_1 + 0.26X_2 + 1.13X_3 + 0.50X_1X_2 + 0.20X_1X_3 + 0.02X_2X_3 - 0.36X_1{}^2 - 0.40X_2{}^2 - 0.36X_3{}^2$$

The ANOVA results of the fitted quadratic polynomial equation are shown in Table 3. The quadratic regression model was highly significant, as was evident from a high model *F*-value of 19.65 and a low probability value at $P < 0.0004$. The Lack of Fit *F*-value of 3.8 was low and its *P*-value (0.1149) was greater than 0.05, implying that the Lack of Fit was statistically insignificant, and the model was not related to the pure error. The determination coefficient $R^2$ of 0.9619 was high, indicating that 96.19% of total variation of the biomass yield could be explained by the model. The adjusted $R^2$ (Adj $R^2$) was in reasonable agreement with $R^2$, indicating a high correlation between the experimental and predicted values. The linear effects of potato ($X_1$) and corn flour ($X_3$) and the interactions of $X_1 \times X_2$ (sucrose) were significant. All quadratic terms

**Table 3. ANOVA results of the fitted quadratic model for the biomass yield of *G. lucidum* in submerged cultures.**

| Source | Sum of squares | DF | Mean square | *F*-value | *Prob>F* |
|---|---|---|---|---|---|
| Model | 18.16 | 9 | 2.02 | 19.65 | 0.0004 |
| $X_1$: Potato | 4.26 | 1 | 4.26 | 41.51 | 0.0004 |
| $X_2$: Sucrose | 0.52 | 1 | 0.52 | 5.07 | 0.0591 |
| $X_3$: Corn flour | 10.22 | 1 | 10.22 | 99.46 | <0.0001 |
| $X_1 \times X_2$ | 1.00 | 1 | 1.00 | 9.74 | 0.0168 |
| $X_1 \times X_3$ | 0.16 | 1 | 0.16 | 1.56 | 0.2521 |
| $X_2 \times X_3$ | 1.600E-003 | 1 | 1.600E-003 | 0.016 | 0.9042 |
| $X_1^2$ | 0.55 | 1 | 0.55 | 5.40 | 0.0531 |
| $X_2^2$ | 0.68 | 1 | 0.68 | 6.66 | 0.0365 |
| $X_3^2$ | 0.55 | 1 | 0.55 | 5.40 | 0.0531 |
| Residual | 0.72 | 7 | 0.10 | | |
| Lack of fit | 0.53 | 3 | 0.18 | 3.80 | 0.1149 |
| Pure error | 0.19 | 4 | 0.047 | | |
| Cor total | 18.88 | 16 | | | |
| $R^2$ | 0.9619 | | | | |
| Adj $R^2$ | 0.9130 | | | | |
| Pred $R^2$ | 0.5336 | | | | |
| Adeq precision | 15.135 | | | | |
| CV | 5.82% | | | | |

($X_1^2$, $X_2^2$, and $X_3^2$) had no significant effect on biomass production. The coefficient of variation (CV) of 5.82% was not high, indicating the accuracy and reliability of the experiments performed.

### Response surface analysis of the independent variables on biomass production

The graphical representations of the fitted model, called response surface curve and contour plots, are illustrated in Fig 3. These graphs demonstrated the individual and interactive effects of the three investigated compositions on biomass production of *G. lucidum* in submerged cultures. The three-dimensional response surface plots (Fig 3A) indicated that high levels of both potato ($X_1$) and sucrose ($X_2$) were favorable for mycelial growth, but the two variables' degree of influence was not high as the slope of the curvature was low. An elliptical nature of contour plots between potato and sucrose indicated that the interactive effect of the two variables ($X_1 X_2$) on biomass production was significant.

Fig 3B shows the combined effects of potato ($X_1$) and corn flour ($X_3$) on biomass production while keeping the third variable ($X_2$, sucrose) at its zero level. The response surface curves display that increasing amount of either potato or corn flour inside the limits of the investigated region can increase biomass yield. The interaction between potato and corn flour ($X_1 X_3$) was negligible, as the shape of contour plots was circular in nature (Fig 3b).

The curvature of response surface between sucrose ($X_2$) and corn flour ($X_3$) in Fig 3C indicated that changing sucrose concentration within the investigated region had little effect on biomass yield. Contrary to this, a small variation in corn flour amount would lead to a considerable impact on the response variable (biomass yield), with the preference at higher concentration. A high biomass production could be obtained at a relatively high level of corn flour (7.5 g/L) when keeping sucrose at the vicinity of its zero level (25.0 g/L). The contour plots between sucrose and corn flour was circular in shape (Fig 3c), implying that there were no significant interactions between these two variables.

Fig 3. Response surface (A, B, and C) and contour (a, b, and c) plots showing the individual and interactive effects of the three medium compositions on biomass production of *G. lucidum* in submerged cultures.

## Validation of the mathematical model

Using the numerical optimization technique provided by the software Design-Expert and based on response surface analysis, the optimal levels of the three investigated compositions for maximal biomass yield were attained as 300 g/L potato, 25 g/L sucrose, and 7.5 g/L corn flour. To validate the predictive model, verification experiments were carried out using the optimized medium. The biomass production reached 7.51 g/L, close to the predicted value of 7.37 g/L, implying the reliability and accuracy of the quadratic polynomial model.

## Bioactive and proximate composition of the cultured mycelia

Medium composition had a significant effect on the content of nutritional and bioactive compositions in the mycelia of several mushroom strains during submerged fermentation [24,25]. Table 4 lists the content of the bioactive and proximate composition of the mycelia cultivated in the optimized and NR media. The crude protein content of the mycelia grown in the optimized medium was 38.71%, lower than that (51.82%) of the mycelia cultured in the NR medium. The fiber content was high, reaching 10.51%, 77.97% higher than that of the mycelia (5.93%) grown in the NR medium. The crude ash content was high, reaching 10.43%, while it was only 5.75% in the NR medium-cultured mycelia. Total carbohydrate content of the mycelia cultured in the optimized medium was 39.03%, higher than that (35.29%) of mycelia harvested from the NR medium. The difference in the crude fat content in the mycelia between the two media was little.

The triterpenoids and polysaccharides are the two most important bioactive constituents in *G. lucidum.* As shown in Table 4, the mycelia cultivated in the optimal medium had a triterpenoid content of 32.44 mg/g, while it was only 20.35 mg/g in the mycelia grown in the NR medium. The intracellular polysaccharide content was high, reaching 3.42%, and 2.03% in the mycelia grown in the NR medium. Clearly, there existed significant differences in the content of triterpenoids and polysaccharides between the mycelia grown in the optimized and NR media.

## Content of amino acid compositions in the cultured mycelia

The content of amino acid compositions of the mycelia cultivated in the optimized medium and the NR medium are listed in Table 5. The total amino acid content in the mycelia cultivated in the optimized medium was 262.4 mg/100g, and the essential amino acid content was 115.4 mg/100 g, while the mycelia grown in the NR medium had much higher values, with the total amino acid content being 337.1 mg/100 g and the essential amino acid content being 139.5 mg/100 g. The content of all examined amino acid compositions in the mycelia were significantly lower than those in the mycelia grown in the NR medium. Among all examined amino acids, the greatest difference in the content was shown by lysine, arginine and glutamate in the mycelia between the two media.

**Table 4. Contents of bioactive and proximate composition in the mycelia of *G. lucidum* grown in the optimized medium and the NR medium.**

| Components | Medium | |
| --- | --- | --- |
| | **Optimized medium** | **NR medium** |
| Moisture (%) | 5.62±0.04a | 5.78±0.04a |
| Crude ash (%) | 10.43±0.33a | 5.75±0.19b |
| Crude fat (%) | 1.32±0.03a | 1.21±0.02b |
| Crude protein (%) | 38.71±0.51b | 51.82±0.64a |
| Crude fiber (%) | 10.51±0.06a | 5.93±0.04b |
| Carbohydrate (%) | 39.03±1.02a | 35.29±0.84b |
| Intracellular polysaccharides (%) | 3.42±0.21a | 2.03±0.11b |
| Triterpenoids (mg/g) | 32.44±2.17a | 20.35±1.56b |

**Table 5. Content of amino acid compositions in the mycelia of *G. lucidum* cultivated in the optimized medium and the NR medium.**

| Amino acids | Content (mg/100 g) | |
|---|---|---|
| | **Optimized medium** | **NR medium** |
| Asparagine | 24.8±0.32b | 32.0±0.78a |
| Threonine* | 13.9±0.21b | 16.6±0.28a |
| Serine | 14.6±0.55b | 19.0±0.42a |
| Glutamate | 37.1±1.25b | 51.6±1.61a |
| Glycine | 15.1±0.34b | 22.1±0.41a |
| Alanine | 18.9±0.27b | 23.2±0.33a |
| Valine* | 23.0±0.36b | 25.3±0.20a |
| Methionine* | 6.9±0.12b | 7.5±0.21a |
| Isoleucine* | 12.4±0.16b | 14.3±0.15a |
| Leucine* | 21.7±0.22b | 25.3±0.31a |
| Tyrosine* | 7.7±0.14b | 8.1±0.20a |
| Phenylalanine* | 12.8±0.32b | 15.1±0.18a |
| Lysine* | 17.0±0.24b | 27.3±0.34a |
| Histidine | 6.5±0.07b | 8.1±0.16a |
| Arginine | 16.8±0.21b | 25.0±0.36a |
| Proline | 13.2±0.14b | 16.6±0.17a |
| Total essential acids | 115.4±2.35b | 139.5±1.69a |
| Total amino acids | 262.4±3.18b | 337.1±3.26a |

*represents essential amino acids.

Data in each row followed by different letters are significantly different ($P < 0.05$).

## Discussion

It is well known that medium is a crucial factor for the growth of microbes and production of useful metabolites during microbial fermentation. As for submerged fermentation of *G. lucidum,* a large number of media have been used, and the compositions used among them varied significantly [9–11]. Presently, a variety of biochemical reagents like yeast extract and peptone, and several common inorganic salts such as $KH_2PO_4$ and $MgSO_4$ are often utilized in medium development for *G. lucidum* fermentation [13,16]. Moreover, many low-cost agricultural products such as corn flour and agricultural by-products like soymeal have also been utilized [23], mainly focusing on achieving high yields of biomass and biologically active substances, while the safety of the mycelia and broth resulting from the bioprocess has often not been taken into consideration. If these fermentation products are directly used in food and health products, it may give rise to some concerns of food safety.

Currently, there are a few numbers of available reports regarding the use of natural fermentative media in submerged fermentation of *G. lucidum.* Sugarcane juice was employed in *G. lucidum* submerged fermentation, and the concentration of proteins and the percentage of total essential amino acids in the fermentation broth were found to be increased, and the broth was developed into a beverage having features of high antioxidant activity and non-toxicity [26]. In a study done by Yang and Zhang [27], soymilk fermented by *G. ludicum* exhibited improved acceptability and health properties. A liquid medium comprising of black soybean and *Astragalus membranaceus* was utilized in *G. lucidum* fermentation and the broth showed high antioxidant activity [28]. To better utilize resources, cheese whey was used as medium for fermentative production of *G. lucidum* mycelia [29]. The pumpkin juice was used as a culture medium in *G. lucidum* fermentation, and its nutritional quality was greatly improved [30]. In the present study, three low-lost agricultural products, i.e., potato,

sucrose, and corn flour, were utilized as medium ingredients to prepare a food-grade medium for growing *G. lucidum* mycelia, and showed excellent performance.

It has been reported that the composition of culture media had a significant effect on the biosynthesis of active metabolites during submerged fermentation of mushrooms like *Antrodia cinnamomea* and *Ganoderma sinense* [22,31]. In the present study, in order to provide a scientific basis for the better utilization of mycelia cultivated in the optimized natural medium, its nutritional and bioactive composition was investigated. The mycelia contained high contents of triterpenoids and polysaccharides, significantly higher than those of the mycelia grown in the NR medium. Hence, the cultured food-grade mycelia can be used as a desirable material for the extraction of triterpenoids (or ganoderic acids) and polysaccharides, which have the great potential for use in drug development. Besides this usage, the mycelia had the potential to be directly processed into capsules as health products because of its high content of bioactive substances.

Dietary fiber is a non-starch polysaccharide that is resistant to digestion and adsorption in the human small intestine, but which can be fermented in the colon. A wide variety of physiological benefits such as blood sugar and cholesterol attenuation and antitumor activity to human body by dietary fiber have been reported [32]. Mushroom is known to be a good source of dietary fiber, and has attracted much attention [33,34]. Dietary polysaccharides of *G. lucidum* had remarkable positive effects on the growth, immune function, and disease resistance in aquatic animals [35]. In the present study, the crude fiber content of the mycelia cultivated in the optimized medium reached 10.51%, significantly higher than that of other studies [36]. The high content of crude fiber implies that the obtained mycelia could be used as a fiber supplement in diet.

The exopolysaccharides (EPS) are often the major target product in submerged fermentation of mushrooms, however, the EPS concentration in broth is generally low. Furthermore, the separation of EPS requires a large amount of organic solvent. Considering these facts, the recovery of this biomacromolecule is uneconomic. As an alternative, the direct utilization of fermentation broth has been attempted. Zhao et al. [37] reported that the incorporation of the mycelial culture filtrate could improve the quality Chinese steamed bread. The fermentation broth of *G. lucidum* as a novel antioxidant and antimicrobial agent and also as an enzyme inhibition agent has been reported [38,39]. In the present study, as the medium ingredients used are all natural, the broth obtained from *G. lucidum* fermentation on the natural medium had characteristics of high safety to human body. Therefore, the fermentation broth holds great promise for use as a low-cost material in various fields such as the food industry.

In conclusion, a natural fermentative medium was developed and optimized for performing *G. lucidum* submerged fermentation. The mycelia obtained on the natural medium after *G. lucidum* fermentation contained high content of bioactive substances, holding the great promise for use in the pharmacological and health industries. Meanwhile, the fermentation broth had the great potential for safe use in various fields such as the food industry. By using the natural medium in *G. lucidum* submerged fermentation, a zero-waste bioprocess could be well established, and its industrial production may be economically feasible.

## Supporting information

**S1 Data. Raw data for figure illustration.**
(XLSX)

## Author contributions

**Conceptualization:** Shizhong Zheng, Shengrong Liu.

**Formal analysis:** Weirui Zhang.

**Funding acquisition:** Shengrong Liu.

**Investigation:** Shizhong Zheng, Lu Zhang, Yihui Yang.

**Methodology:** Qi Wei, Qianhui Huang.

**Project administration:** Shengrong Liu.

**Supervision:** Shizhong Zheng, Shengrong Liu.

**Writing – original draft:** Shizhong Zheng, Lu Zhang.

**Writing – review & editing:** Shizhong Zheng, Shengrong Liu.

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
