## [Decision Letter · Decision Letter 0]

16 Sep 2025

Dear Dr. Liu,

Thank you for submitting your manuscript to PLOS ONE. After careful consideration, we feel that it has merit but does not fully meet PLOS ONE’s publication criteria as it currently stands. Therefore, we invite you to submit a revised version of the manuscript that addresses the points raised during the review process.

We look forward to receiving your revised manuscript.

Kind regards,

Shwet Kamal, Ph.D

Academic Editor

PLOS ONE

Journal Requirements:

“This work was supported by the financial support of Fujian Provincial Department of Science and Technology (2021I0046).”

3. We note that your Data Availability Statement is currently as follows:

“All relevant data are within the manuscript and its Supporting Information files.”

Additional Editor Comments:

The Manuscript "Optimization of a natural fermentative medium for submerged mycelial culture of Ganoderma lucidum and the nutritional and bioactive composition of the cultured food-grade mycelia' is a nice study. However, the manuscript requires some corrections suggested in the review track mode by the reviewrs, and also follow the grammers and language editing for better shaping the manuscript.

Reviewer #1:

The author presented the paper on Optimization of a natural fermentative medium for submerged mycelial culture of Ganoderma lucidum and the nutritional and bioactive composition of the cultured food-grade mycelia in a good manner. However, the manuscript requires some corrections suggested in the review track mode, and also follow the grammers and language editing for better shaping the manuscript.

Reviewer #2:

Authors have done nice work on submerged mycelial culture of Ganoderma lucidum and nutritional and bioactive composition of mycelia. Manuscript is very well written and concluded. Some miner changes are there which are highlighted in the manuscript. Those may done in the manuscript for the improvement.

Reviewer's Responses to Questions

**Comments to the Author**

1. Is the manuscript technically sound, and do the data support the conclusions?

Reviewer #1: Yes

Reviewer #2: Yes

2. Has the statistical analysis been performed appropriately and rigorously?

Reviewer #1: Yes

Reviewer #2: Yes

3. Have the authors made all data underlying the findings in their manuscript fully available?

Reviewer #1: Yes

Reviewer #2: Yes

4. Is the manuscript presented in an intelligible fashion and written in standard English?

Reviewer #1: Yes

Reviewer #2: Yes

Reviewer #1: The author presented the paper on Optimization of a natural fermentative medium for submerged mycelial culture of Ganoderma lucidum and the nutritional and bioactive composition of the cultured food-grade mycelia in a good manner. However, the manuscript requires some corrections suggested in the review track mode, and also follow the grammers and language editing for better shaping the manuscript.

Reviewer #2: Authors have done nice work on submerged mycelial culture of Ganoderma lucidum and nutritional and bioactive composition of mycelia. Manuscript is very well written and concluded. Some miner changes are there which are highlighted in the manuscript. Those may done in the manuscript for the improvment.

**Do you want your identity to be public for this peer review?** For information about this choice, including consent withdrawal, please see our Privacy Policy

Reviewer #1: No

Reviewer #2: No

---

## [Author Response · Author response to Decision Letter 1]

1 Oct 2025

Responses to the reviewer’s comments

Manuscript number: PONE-D-25-34815

Title: Optimization of a natural fermentative medium for submerged mycelial culture of Ganoderma lucidum and the nutritional and bioactive composition of the cultured food-grade mycelia

Dear Editors,

We appreciate your consideration and giving us the opportunity to revise our manuscript. The manuscript has been revised according to the comments and suggestions, and all problems required have been addressed. Revised portion are marked in red in the paper. Point-by-point responses to these comments are showed below in this letter.

Sincerely yours,

Shizhong Zheng, Lu Zhang, Yihui Yang, Shengrong Liu, Qi Wei, Qianhui Huang, Weirui Zhang

Editors: Some douts, suggestions and recommendation

Comment 1: The Manuscript “Optimization of a natural fermentative medium for submerged mycelial culture of Ganoderma lucidum and the nutritional and bioactive composition of the cultured food-grade mycelia” is a nice study. However, the manuscript requires some corrections suggested in the review track mode by the reviewrs, and also follow the grammers and language editing for better shaping the manuscript.

Response: Thanks. We have carefully checked the manuscript.

Reviewer 1#: Some douts, suggestions and recommendations

Comment 1: Line 127-128 “Why corn flour was using. Is there any scientific evidence for using the corn flour”.

Response: Accept. corn flour is widely used in the fermentation industry, and for culturing mushroom strains. We have cited several references to support our choice for use of corn flour. See in the revision.

Comment 2: Line 181: “Have you seen the proximate analysis for the same under basic media”.

Response: Your suggestion is greatly appreciated. As the difference in the medium nutrition level especially nitrogen source between the optimized medium and the basic medium was not large, we selected a nitrogen-rich medium as a reference for comparison. The proximate composition analysis of the cultured mycelia between the two media was carried out, and there were significant differences in chemical compositions of the mycelia between the two media.

Comment 3: Line 182: “What is the added advantage of your supplementations”.

Response: Thanks. In mushroom submerged liquid fermentation, the mycelial biomass is an important index. In the present research, the addition of corn flour was adopted mainly for promoting mycelial growth.

Reviewer 2#: Some douts, suggestions and recommendations

Comment 1: Line 22: “What is the need of this research? Explain it is to develop for what?”

Response: Accept. The objective of the present study has been added in the revised manuscript.

Comment 2: Line 66-68: This has to be reflect in abstract session.

Response: Thanks. This part is cited from the reference, it was not appropriate to remove to the abstract session.

Comment 3: Line 303: “Usually, PDA or PDB will be the basic media for mushroom fungal culture. Have you check for PDA to compare the optimized one?”.

Response: Thanks. PDB is often used as a basic medium for mushroom fungal culture. In the present research, after the medium optimization, the biomass yield was greatly enhanced as compared with PDB medium. On the other hand, the difference in the medium nutrition level especially nitrogen source between the optimized medium and PDB was not large, we therefore selected a nitrogen-rich medium as a reference for comparison.

List of the changes in revision

Abstract

1. Line 21-22 of page 2, “with healthy-promoting benefits and pharmacological properties” was changed as “with health-promoting and pharmacological properties”.

2. Line 22-23 of page 2, “To safely utilize the mycelia and broth of G. lucidum by submerged fermentation” was changed as “The present study aimed to provide a liquid medium in G. lucidum submerged fermentation suitable for safely utilizing the mycelia and broth and as a solution to potentially improve the process’ economy.”.

3. Line 23-25 of page 2, “a natural fermentative medium was formulated and optimized by the one-variable-at-a-time approach and response surface methodology (RSM)” was changed as “For this, a natural medium was formulated and optimized.”.

4. Line 26-27 of page 2, “were compared” was changed as “were also determined”.

5. Line 27 of page 2, “ tested” was inserted .

6. Line 28 of page 2, “additional” was deleted.

7. Line 29-31 of page 2, “A further optimization by RSM indicated that the optimal medium was comprised of 300 g/L potato (use its extract), 25 g/L sucrose, and 7.5 g/L corn flour” was changed as “A further study by response surface methodology obtained an optimal medium comprising of 300 g/L potato (use its extract), 25 g/L sucrose, and 7.5 g/L corn flour”.

8. Line 31 of page 2, “With” was changed as “using”.

9. Line 32-34 of page 2, “The crude ash (10.43%) and fiber (10.51%) content in the mycelia cultivated with the optimal medium was much higher than that (5.75% and 5.93%) of the mycelia grown in the NR medium” was changed as “The crude ash (10.43%) and fiber (10.51%) content of the mycelia cultivated in the optimal medium was higher than those (ash 5.75% and fiber 5.93%) of the mycelia grown in the NR medium”.

10. Line 35 of page 2, “the latter” was changed as “the mycelia cultivated in the NR medium”.

11. Line 36-38 of page 2, “The mycelia grown the optimized medium had 32.24 mg/g triterpenoids and 3.42% intracellular polysaccharides, significantly higher than that (20.35 mg/g and 2.03%) of the mycelia by growing in the NR medium” was changed as “The mycelia had the content of triterpenoids at 32.24 mg/g and intracellular polysaccharides at 3.42%, higher than those (triterpenoids 20.35 mg/g and intracellular polysaccharides 2.03%) of the mycelia cultivated with the NR medium”.

12. Line 38-40 of 2, “These findings proved that the food-grade cultured mycelia had the potential for use in the pharmacological industry because of its high content of triterpenoids and polysaccharides” was changed as “These findings indicated that the cultured food-grade mycelia had the potential for use in the pharmacological industry”.

13. Line 40-41 of page 2, “and as a dietary supplement in the food and healthy products” was deleted.

14. Line 41 of page 2, “Meanwhile, the broth had the potential for safe use in foods due to free of chemicals” was inserted.

15. Line 41-44 of page 2, “The present study provides a reference for improving the economic benefits of G. lucidum submerged culture by using a natural medium, providing the condition for the safe utilization of all fermentation products, and thus its large-scale industrial fermentation may be economically feasible” was changed as “The present study provides a reference for improving the economy of G. lucidum submerged culture by using a natural medium, and its industrial fermentation may be economically viable”.

16. Line 44 of page 2, “Keywords: Lingzhi, medium optimization, mycelium, triterpenoids, polysaccharides” was inserted.

Introduction

1. Line 47 of page 3, “famous” was inserted.

2. Line 55 of page 3, “compounds” was deleted.

3. Line 56 of page 3, “ in this fungus” was inserted.

4. Line 57 of page 3, “To meet the high demand for high-quality G. lucidum in market” was changed as “To meet the great demand for high-quality G. lucidum products in market”.

5. Line 58 of page 3, “solid” was inserted.

6. Line 58 of page 3, “is a widely adopted method” was changed as “widely adopted”.

7. Line 59 of page 3, “With the fruiting bodies as raw materials” was changed as “With the fruiting bodies and spores as raw materials”.

8. Line 59 of page 3, “wide” was added.

9. Line 60-61 of page 3, “could be manufactured” was changed as “have been developed and manufactured ”.

10. Line 61 of page 3, “solid” was inserted.

11. Line 62 of page 3, “some” was changed as “several”.

12. Line 62-63 of page 3, “there are several disadvantages associated with solid cultivation” was changed as “it should be pointed out that the solid cultivation of G. lucidum has several disadvantages”.

13. Line 68 of page 3, “in the form of mycelia and bioactive metabolites” was inserted.

14. Line 68-70 of page 3-4, “In submerged fermentation of mushrooms, the mycelia and small quantities of several useful metabolites could be obtained” deleted.

15. Line 70 of page 4, “healthy-promoting” was changed as “health-promoting”.

16. Line 71-72 of page 4, “considerable attention has been paid to its submerged cultures for the production of mycelia, polysaccharides and triterpenoids” was changed as “its submerged culture has received considerable attention, mainly for the production of mycelia, polysaccharides and triterpenoids”.

17. Line 75 of page 4, “development of solid seed” was changed as “utilization of solid seed”.

18. Line 76 of page 4, “some improvements” was revised as “significant improvements”.

19. Line 76 of page 4, “However” was deleted.

20. Line 77-79 of page 4, “the productivity of mycelia and bioactive metabolites are still low in submerged fermentation of G. lucidum, making the process industrially unattractive” was revised as “the productivity of both mycelia and bioactive metabolites are still low, making the submerged fermentation process of G. lucidum industrially unattractive”.

21. Line 79-82 of page 4, “Increasing product yields and developing novel production systems by solving the existing problems associated with submerged fermentation technique of mushrooms are considered to be crucial for its success on a commercial scale” was changed as “Another obstacle is that there are many technical and engineering problems associated with G. lucidum submerged fermentation in bioreactors”.

22. Line 82 of page 4, “In submerged fermentation of mushrooms” was deleted.

23. Line 82-84 of page 4, “utilizing all the fermentation products including mycelia and culture broth may be another solution to improve bioprocess economy” was changed as “Under these circumstances, the effective utilization of all the fermentation products including mycelia and culture broth may be a reasonable solution to improve the bioprocess’ economy”.

24. Line 84-85 of page 4, “but the effect of medium compositions on the safety of fermentation products has not yet been considered when they were selected” was deleted.

25. Line 86-88 of page 4, “to develop a natural medium for submerged mycelial culture of G. lucidum from a perspective for the safe utilization of fermentation products (mycelia and culture broth)” was revised as “to provide a desirable medium for submerged mycelial culture of G. lucidum suitable for the safe utilization of its fermentation products (mycelia and culture broth)” .

26. Line 89 of page 4, “then” was inserted.

27. Line 91 of page 4, “Furthermore” was inserted.

28. Line 91 of page 4, “better” was added.

29. Line 92 of page 4, “harvested from the developed natural medium” was deleted.

30. Line 93 of page 4, “in the mycelia” was deleted.

31. Line 93-96 of page 4-5, “The present study provides a new perspective for improving the economic benefits of G. lucidum submerged fermentation by using a natural medium, ultimately leading to the safe utilization of all fermentation products” was revised as “The present study provides a new perspective to perform G. lucidum submerged fermentation by using a natural medium for the safe utilization of all the fermentation products, thus could potentially improve the economy of process”.

32. Line 96 of page 5, “and thus its large-scale industrial fermentation may be feasible” was revised as “making its large-scale industrial fermentation economically feasible”.

Materials and methods

1. Line 98 of page 5, “and medium” was deleted.

2. Line 100 of page 5, “used in this study” was changed as “throughout this study”.

3. Line 105 of page 5, “For comparison”was inserted”.

4. Line 106 of page 5, “Nutrient-rich (NR) medium was comprised of ” was changed as “a nutrient-rich (NR) medium was used, which was comprised of”.

5. Line 109 of page 5, “actively” was inserted.

6. Line 112 of page 5, “mycelial” was deleted.

7. Line 116-117 of page 5, “in subsequent studies” was deleted.

8. Line 120 of page 6, “The inoculation of liquid seed was set at 10% (v/v)” was changed as “ The inoculation level of liquid seed was all set at 10% (v/v)”.

9. Line 121-122 of page 6, “The samples were taken for harvesting mycelia” was changed as “The samples were taken for estimating mycelial biomass”.

10. Line 123 of page 6, “One-variable-at-one-time experiments” was changed as “One-variable-at-one-time experimental design”.

11. Line 124 of page 6, “in G. lucidum submerged cultures” was inserted.

12. Line 124-125 of page 6, “To select a suitable carbon source for mycelial yield of G. lucidum in submerged cultures” was changed as “To select a suitable carbon source for mycelial growth”.

13. Line 125 of page 6, “the glucose in the medium” was changed as “the glucose in the basic medium”.

14. Line 126 of page 6, “At the end of submerged cultivation, the biomass production was measured, and then a desirable carbon source was selected” was inserted.

15. Line 126-127 of page 6, “After this” was changed as “After the screening”.

16. Line 127 of page 6, “easily available” was inserted.

17. Line 128 of page 6, “with the optimal carbon source” was changed as “with the selected carbon source”.

18. Line 129 of page 6, “based on biomass yield” was inserted.

19. Line 141 of page 6, “in submerged cultures” was inserted.

20. Line 141-142 of page 6-7, “Corn powder” was changed as “Corn flour”.

21. Line 142 of page 7, “2” was changed as “2.5”.

22. Line 143 of page 7, “The matrix of BBD design” was revised as “The matrix of BBD”.

23. Line 145 of page 7, “The submerged culture experiments were performed as previously described” was inserted.

24. Line 153 of page 7, “The statistical significance of the equation” was revised as “The significance of the obtained regression model”.

25. Line 158 of page 7, “indicates” was revised as “indicate”.

26. Line 159 of page 7, “were statistically significant” was changed as “were considered statistically significant”.

27. Line 162 of page 7, “the type of interactions between the variables” was changed as “the type of interaction between the two variables”.

28. Line 169 of page 8, “For production of mycelia” was changed as “For the production of mycelia”.

29. Line 169 of page 8, “in flasks” was changed as “in 250 mL flasks”.

30. Line 170 of page 8, “Submerged cultivation was also performed using the NR medium” was changed as “For comparison, the NR medium was also tested in the fermentation”.

31. Line 171-172 of page 8, “The mycelial cultures were filtered for harvesting mycelia, and washed with distilled water” was changed as “The mycelial cultures were filtered with gauze and completely washed with distilled water”.

32. Line 173 of page 8, “were used for analysis” was changed as “ were used for the chemical analysis”.

33. Line 175-178 of page 8, “three independent culture flasks” was changed as “three independent flask cultures”.

34. Line 178 of page 8, “The dry mycelial weight” was changed as “The weight of dry mycelia”.

35. Line 187-188 of page 9, “by subtracting the contents of ash, fat, and protein from 100” was changed as “ by subtracting the percentage contents of ash, fat, and protein from 100%”.

36. Line 188-189 of page 9, “Amino acid analysis was carried out after hydrolysis of the mycelia with 6 mol/L HCl for 24 h at 110C using an amino acid analyzer” was changed as “ Amino acid analysis was carried out using an amino acid analyzer after hydrolysis of the mycelia with 6 mol/L HCl for 24 h at 110C”.

37. Line 199-200 of page 9, “The powder (100 mg) was extracted twice in 5 mL 80% ethanol in one week” was changed as “The powder (100 mg) was extracted twice in one week, each in 5 mL 80% ethanol”.

38. Line 201 of page 9, “for analysis” was revised as “in the reaction”.

39. Line 205-206 of page 9, “by Duncan’s multiple range tests (P < 0.05)” was revised as “by Duncan’s multiple range tests and Student’s t-test at P < 0.05”.

Results

1. Line 212 of page 10, “Among the tested carbon sources,” was changed as “Among various tested carbon sources”.

2. Line 213 of page 10, “and maltose, glucose and fructose s

---

## [Editor Report · Decision Letter 1]

11 Nov 2025

Optimization of a natural fermentative medium for submerged mycelial culture of Ganoderma lucidum and the nutritional and bioactive composition of the cultured food-grade mycelia

PONE-D-25-34815R1

Dear Dr. Liu,

We’re pleased to inform you that your manuscript has been judged scientifically suitable for publication and will be formally accepted for publication once it meets all outstanding technical requirements.

Kind regards,

Shwet Kamal, Ph.D

Academic Editor

PLOS ONE

Additional Editor Comments (optional):

"Optimization of a natural fermentative medium for submerged mycelial culture of Ganoderma lucidum and the nutritional and bioactive composition of the cultured food-grade mycelia" is a nice study executed in a good manner. The reviewrs suggested minor revisions which the authors have done and responded well. Some changes are required in the manuscript related to spellings and contexts, which is now marked in the manuscript. After corrections, the manuscript can be accepted for publication.
---

## [Editor Report · Acceptance letter]

PONE-D-25-34815R1

PLOS One

Dear Dr. Liu,

I'm pleased to inform you that your manuscript has been deemed suitable for publication in PLOS One. Congratulations! Your manuscript is now being handed over to our production team.

Kind regards,

on behalf of

Dr. Shwet Kamal

Academic Editor

PLOS One